# Chiral excitonic order from twofold van Hove singularities in kagome metals

Harley D. Scammell [1,2] ✉, Julian Ingham [3] ✉, Tommy Li[4] & Oleg P. Sushkov[1,2]

Recent experiments on kagome metals $AV_3Sb_5$ (A=K,Rb,Cs) identify twofold van Hove singularities (TvHS) with opposite concavity near the Fermi energy, generating two approximately hexagonal Fermi surfaces – one electron-like and the other hole-like. Here we propose that a TvHS generates a novel time-reversal symmetry breaking excitonic order – arising due to bound pairs of electrons and holes located at opposite concavity van Hove singularities. We introduce a minimal model for the TvHS and investigate interaction induced many-body instabilities via the perturbative renormalisation group technique and a free energy analysis. Specialising to parameters appropriate for the kagome metals $AV_3Sb_5$, we construct a phase diagram comprising chiral excitons, charge density wave and a region of coexistence. We propose this as an explanation of a diverse range of experimental observations in $AV_3Sb_5$. Notably, the chiral excitonic state gives rise to a quantum anomalous Hall conductance, providing an appealing interpretation of the observed anomalous Hall effect in kagome metals. Possible alternative realisations of the TvHS mechanism in bilayer materials are also discussed. We suggest that TvHS open up interesting possibilities for correlated phases, enriching the set of competing ground states to include excitonic order.

Kagome systems have been a major focus of theoretical and experimental investigation; due to their ability to realise Dirac points, flat bands and van Hove singularities, they have been predicted to host a range of novel correlated phases of matter[1–9]. Recently, a new class of materials $AV_3Sb_5$ (A=K,Rb,Cs) have attracted a great deal of attention due to their demonstration of unconventional superconductivity alongside competing density wave order, spatially modulated superconducting order and possible signatures of Majorana states in superconducting vortices[10–43]. Unusually, the materials exhibit time-reversal symmetry breaking with an anomalous Hall conductivity in spite of the absence of magnetic ordering; the origins and relationship between superconductivity, competing order, and the anomalous Hall effect remain an open question.

The materials consist of a stack of two dimensional layers—a kagome lattice of vanadium and antimony alternating with a hexagonal lattice of antimony and triangular lattice of the alkali metal K/Rb/Cs— with electrical transport predominantly in-plane, as demonstrated by the large ratio between the out-of- and in-plane resistivity $R_c/R_{ab} \approx 600$. The Fermi surface of these materials consists of several distinct contours, including nearly circular contours centred at the $\Gamma$ and $K$ points as well as two approximately hexagonal contours[44]. Systems with hexagonal Fermi surfaces, corresponding to saddle-points in the electronic dispersion, have been predicted to give rise to chiral superconductivity and competing density wave order, due to the effects of Fermi surface nesting[45]. However, ARPES and DFT results reveal that the hexagonal Fermi surfaces in the vanadium metals exhibit an unusual feature—twofold van Hove singularities (TvHS), for which the saddle-points at each Fermi surface possess opposite concavity, resulting in one electron-like Fermi surface and one hole-like Fermi surface[39].

[1]School of Physics, University of New South Wales, Sydney, NSW 2052, Australia. [2]Australian Research Council Centre of Excellence in Future Low-Energy Electronics Technologies, University of New South Wales, Sydney, NSW 2052, Australia. [3]Physics Department, Boston University, Commonwealth Avenue, Boston, MA 02215, USA. [4]Dahlem Center for Complex Quantum Systems and Fachbereich Physik, Freie Universität Berlin, Arnimallee 14, 14195 Berlin, Germany. ✉e-mail: harleyscammell@gmail.com; jingham@bu.edu

We argue that doping a system to a TvHS has an ineluctable influence on the low-energy physics. A single vHS results in a tendency towards density wave ordering and superconductivity. The appearance of TvHS introduces an additional tendency towards excitonic order—corresponding to a condensation of electron-hole pairs—owing to the coupling between an electron-like and hole-like Fermi surface. We introduce a low-energy model, which incorporates the TvHS—featuring an electron-like and hole-like Fermi surface, each doped near their respective vHS. To understand the interplay and competition between the various many-body instabilities, we employ the perturbative renormalisation group (RG) method to determine the dominant ground state order[45–49], complemented by a Landau-Ginzburg free energy analysis of competing ground states. A chiral excitonic order naturally emerges, which breaks time-reversal symmetry and exhibits a quantum anomalous Hall effect. The chiral excitonic state appears as a generic weak coupling instability, but explicit modelling for $AV_3Sb_5$ suggests these materials exist in an intermediate coupling regime; guided by ab initio results we generate a phase diagram featuring charge density wave order, chiral excitonic order and a region of coexistence. We suggest that the phenomenology encompassed by the TvHS model accounts for key features observed in the vanadium-based kagome metals, and could further motivate TvHS engineering in van der Waals heterostructures and bilayer materials.

## Results

### Tight-binding Hamiltonians with TvHS

A TvHS consists of two Fermi surfaces with opposite concavity vHS so that one surface is electron-like and the other hole-like—e.g., arising from doping near the $M$-point of a 2D hexagonal Brillouin zone as shown in Fig 1a. The opposing concavities of the respective saddle-points can be seen from the colour plot in Figs. 1c; going from outside to inside the hexagonal Fermi surface, the energy changes sign, but the sign change is opposite for the two Fermi surfaces. The two Fermi surfaces may arise due to two hexagonal (honeycomb or kagome)

bilayers, or a single layer with two sets of orbitals—the latter case is the origin of the TvHS in vanadium metals $AV_3Sb_5$.

To be explicit, we will introduce a particular lattice model realisation of a TvHS. A tight-binding model of a kagome monolayer with two sets of orbitals that has been used to describe $AV_3Sb_5$ is given by

$$H_{tb} = -\sum_{\langle i,j \rangle, \nu} t^\nu a^\dagger_{i,\nu} a_{j,\nu} - \sum_{i,\nu} \epsilon^\nu a^\dagger_{i,\nu} a_{i,\nu}, \qquad (1)$$

where $a^\dagger_{i,\nu}$ creates fermions on site $i$ and in orbital $\nu = c, d$. The differing orbital potentials, $\epsilon^c - \epsilon^d \approx t^c + t^d$ shift the energies of the two bands, aligning their valence and conduction bands and resulting in a TvHS.

The bandstructure of a TvHS can be realised in both honeycomb and kagome systems (we discuss alternative tight-binding models in the Supplementary Material). However, the orbital content of the wavefunction at the $M$-points is qualitatively different in these two cases. For honeycomb, with two sublattices, the wavefunction at the $M$-points has equal support on both sublattices. Meanwhile, in kagome systems the wavefunction near the $M$-points exhibits the so-called 'sublattice interference effect'[8]: at a given $M$-point, the conduction band wavefunctions have support only on one sublattice and are referred to as $p$-type (owing to their 'pure' sublattice composition) while the valence band wavefunctions have support on the other two sublattices and are referred to as $m$-type (due to their 'mixed' sublattice composition). The sublattice structure has important consequences when considering interaction effects, as we discuss below.

### Patch model

The problem of interaction driven instabilities on a single hexagonal Fermi surface (i.e., single vHS) has been previously studied using a three patch model[45], whereby the full Brillouin zone is restricted to three momentum space patches near the vHS at the $M$-points, since they dominate the density of states. Following this approach, we define a three patch model and further introduce a flavour degree of freedom

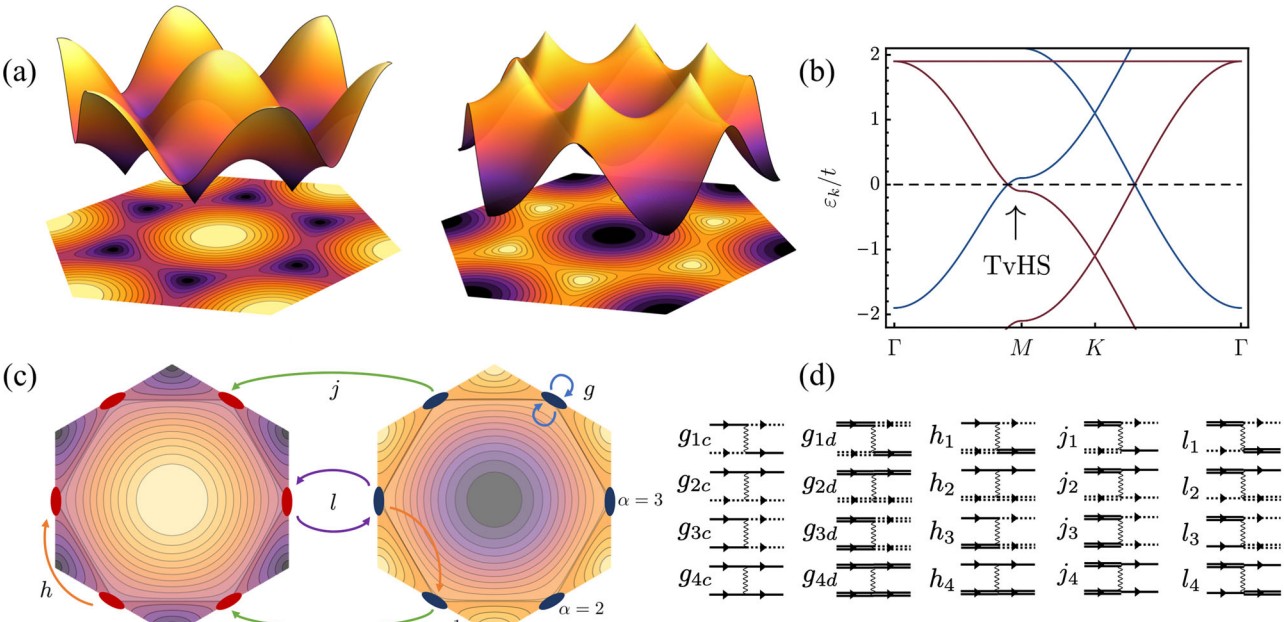

**Fig. 1 | Theoretical model. a** Dispersion plots for a hexagonal tight-binding model, featuring saddle-points with opposite concavity. Dark (light) colours represent negative (positive) energy states relative to the saddle point. **b** Bandstructure for (1), with $t^c = t^d = t$, $\epsilon^c/t = 2.1$, $\epsilon^d/t = -0.1$, demonstrating positive and negative concavity van Hove singularities near the Fermi level. Bands arising from different orbitals are coloured blue and red, respectively contributing an $m$- and $p$-type vHS near the Fermi level. **c** Representative interaction processes from each of the

classes $V_g, V_h, V_j, V_l$ (3), accounting for scattering processes on or between the patches $\alpha = 1, 2, 3$, and flavours (left and right hexagons), illustrated on a contour plot of the energy in the first Brillouin zone. The opposite concavities of the saddle-points can be seen by the opposite sign of the energies inside the Fermi surface. **d** Feynman diagrams representing the full set of allowed scattering processes. Double/single lines denote fermions from the two distinct Fermi surfaces, while dashed/solid lines represent fermions at different patches.

to account for the two opposite concavity Fermi surfaces—fermions of one flavour (created by $c^\dagger$) are electron-like, while fermions of the other flavour (created by $d^\dagger$) are hole-like,

$$H_0 = \sum_{\boldsymbol{k},\alpha}^{\Lambda} (\varepsilon^c_{\boldsymbol{k},\alpha} - \mu) c^\dagger_{\boldsymbol{k},\alpha} c_{\boldsymbol{k},\alpha} - (\varepsilon^d_{\boldsymbol{k},\alpha} + \mu) d^\dagger_{\boldsymbol{k},\alpha} d_{\boldsymbol{k},\alpha}. \qquad (2)$$

The patch index $\alpha = 1, 2, 3$ indicates a fermionic excitation within a cutoff $\Lambda$ of the momentum $\boldsymbol{M}_\alpha$. Setting $\mu = 0$ corresponds to doping exactly to the TvHS. The opposite concavity is encoded by the relative minus sign between the $c$ and $d$ dispersions. The TvHS should be contrasted with the problem of fixed concavity vHS with multiple flavours[50]—the opposite concavity of the two vHS plays a fundamental role in the interaction physics. The patch dispersion take the saddle point form $\varepsilon^\nu_{\boldsymbol{k},1} = \frac{1}{2} t^\nu (k_x^2 + \sqrt{3} k_x k_y)$, $\varepsilon^\nu_{\boldsymbol{k},2} = \frac{1}{4} t^\nu (-k_x^2 + 3k_y^2)$, $\varepsilon^\nu_{\boldsymbol{k},3} = \frac{1}{2} t^\nu (k_x^2 - \sqrt{3} k_x k_y)$, where $t^\nu$ is a characteristic energy scale, and equals the nearest neighbour hopping of the $\nu$-fermions ($\nu = c, d$) in the simple tight-binding model. Fermions at patches $\alpha \neq \beta$ are connected by the nesting vector $\boldsymbol{Q}_{\alpha\beta} = \boldsymbol{M}_\alpha - \boldsymbol{M}_\beta$, for which $\varepsilon^\nu_{\boldsymbol{k}+\boldsymbol{Q}_{\alpha\beta},\beta} \approx -\varepsilon^\nu_{\boldsymbol{k},\alpha}$.

Making contact with ab initio results for AV$_3$Sb$_5$, the $c$- ($p$-type) and $d$- ($m$-type) fermions arise from the vanadium $d_{yz}$ and $d_{xz}$ orbitals respectively, and have $t^c \approx 0.5$ eV, $t^d \approx 1$ eV[51]. In the patch model, this sets $t^c/t^d = \kappa = 2$. For completeness we will analyse both $\kappa = 1$ and $\kappa = 2$. It is known from ARPES that the $c$-band vHS is near-perfectly nested, while the $d$-band vHS exhibits quartic corrections[39]. Close to the $M$-point these corrections are subdominant to the quadratic part of the dispersion, and hence only influence the ultraviolet behaviour of the theory, near the cutoff $\Lambda \approx 0.5$ eV. Since our analysis probes infrared scales far below $\Lambda$, it is well-justified to ignore the quartic corrections.

Below we will analyse three distinct cases: (i) honeycomb systems, for which the sublattice support on the two-flavour vHS is the same, in the particle-hole symmetric limit $\kappa = 1$; (ii) kagome systems, in which the two-flavour vHS have different sublattice support, i.e., $m$- and $p$-type, with $\kappa = 1$; (iii) kagome systems with $\kappa = 2$, which we have argued to describe kagome metals AV$_3$Sb$_5$.

## Interactions

We now consider the possible couplings between the fermions. Owing to the large density of states near the TvHS the Coulomb repulsion is expected to be strongly screened and we therefore model the interactions as short-ranged. The most general set of interactions between patches/flavours allowed by momentum conservation are

$$V = \frac{1}{2} \sum_{\alpha,\beta} \left[ V_{g,\nu} + V_h + V_j + V_l \right] \qquad (3)$$

where $V_{g,\nu}$ are intraflavour couplings, $V_h$ are interflavour density-density couplings, $V_j$ are flavour pair hopping, and $V_l$ are flavour exchange couplings, resulting in 20 independent interactions. A schematic illustration of the $g, h, j, l$ couplings, as well as their representation in terms of Feynman diagrams, is shown in Fig. 1c, d. Additional details are found in the Supplementary Material.

In the kagome case, projecting the sublattice wavefunctions onto the Coulomb interaction results in different intraflavour couplings depending on whether the flavour has pure or mixed sublattice structure, a manifestation of the sublattice interference effect in kagome patch models. We have therefore allowed for different couplings $V_{g,\nu}$ on each flavour. Performing this projection explicitly and using the calculations of ref. [51] gives the estimates of the bare coupling values shown in Table 1. The values taken from ref. [51] are defined at the lattice scale; using these as input to our effective theory neglects the renormalisation flow between the lattice scale and $\Lambda$. The sublattice interference effect has crucial consequences for the bare couplings: for instance, on a $p$-type vHS, the wavefunctions at different patches

**Table 1 | Estimates of the bare coupling values in AV$_3$Sb$_5$**

| | $g_{i,c}$ | $g_{i,d}$ | $h_i$ | $j_i$ | $l_i$ |
|---|---|---|---|---|---|
| $i = 1$ | 0 | $\frac{1}{4}(U+V)$ | 0 | 0 | $\frac{1}{2}J$ |
| $i = 2$ | $V$ | $\frac{1}{4}U+V$ | $\frac{1}{2}U'+V$ | 0 | 0 |
| $i = 3$ | 0 | $\frac{1}{4}(U+V)$ | 0 | $\frac{1}{2}J'$ | 0 |
| $i = 4$ | $U+V$ | $\frac{1}{2}U+V$ | $V$ | 0 | 0 |

Projecting the pure and mixed sublattice form factors onto the cRPA results of ref. [51] results in the below values, where the intra-orbital, inter-orbital, Hund's, pair hopping, and nearest neighbour repulsions are $U = 1$–$2$ eV, with $U' = 0.8U$, $J = J' = 0.1U$ and $V = 0.3U$

are orthogonal, and so the interpatch Coulomb repulsion is suppressed, resulting in $g_{1,c} = g_{3,c} = 0$. Thus, any attractive interactions present in the system, for e.g., due to phonons, immediately result in attractive couplings.

In the honeycomb case, the orbital form factors are the same for both flavours and so we expect $V_{g,c} \approx V_{g,d}$. This reduces the number of independent coupling constants from 20 to 16. We shall present results for both models below.

## Instabilities

Considering the interacting Hamiltonian,

$$H = H_0 + V, \qquad (4)$$

we determine which instabilities arise within the framework of RG. The instability of the metallic phase and onset of an ordered ground state is signalled by the susceptibility of the associated order parameter: the strongest ordering tendencies are those with most divergent susceptibility. In the case of a nested Fermi surface, a density wave instability arises because the nesting condition $\varepsilon_{\boldsymbol{p}} \approx -\varepsilon_{\boldsymbol{p}+\boldsymbol{Q}}$, implies the total energy of a particle with momentum $\boldsymbol{p}$ and hole with momentum $\boldsymbol{p} + \boldsymbol{Q}$ is approximately zero. Similarly, the energy of an electron and a hole at the TvHS is $\varepsilon^c_{\boldsymbol{p}} + \varepsilon^d_{\boldsymbol{p}} \approx 0$. Without including interactions, it costs zero energy to create either of these particle-hole states, and hence, for an arbitrarily small attraction between particles and holes the system becomes unstable to lowering its energy by spontaneously creating many such pairs, analogous to the usual superconducting instability. The RG method provides an unbiased approach to study competing orders on an equal footing, by resumming the logarithmically divergent corrections to the bare couplings and determining which ordering tendency dominates[45–49].

In Table 2, we enumerate the ordered states which naturally arise in the TvHS model, i.e., those with nesting tendencies. The first three—CDW, SDW and SC—occur in the case of a single vHS. The next three—singlet and triplet excitonic order, as well as interflavour pair density wave (PDW)—are new instabilities introduced by the TvHS.

## RG analysis

We turn now to the RG treatment which identifies the leading instabilities, i.e., the dominant ground states in Table 2. Firstly, we compute the leading $\log^2$ corrections to the bare couplings defined in (3). The equations define how the couplings evolve with the RG time $t$, which is a proxy for the energy scale; here $t \to \infty$ corresponds to taking $T \to 0$. The full RG equations for our model are lengthy, since they involve twenty independent interaction constants (Fig. 1d), so we state their general form here and reserve explicit expressions for the Supplementary Material. The RG equations describing the flow of the couplings $g_i, h_i, j_i, l_i$ (where $i = 1, 2, 3, 4$) take the form

$$\frac{\partial}{\partial t} g_{i,\nu} = \beta_{g_{i,\nu}}(g, j, h, l), \qquad \frac{\partial}{\partial t} h_i = \beta_{h_i}(g, j, h, l),$$
$$\frac{\partial}{\partial t} j_i = \beta_{j_i}(g, j, h, l), \qquad \frac{\partial}{\partial t} l_i = \beta_{l_i}(g, j, h, l), \qquad (5)$$

**Table 2 | The leading ordered states**

|  | Structure | vHS | TvHS |
|---|---|---|---|
| CDW | $\mathcal{C}_{\alpha\beta\nu} = \langle \psi_\alpha^\dagger \sigma_\nu \psi_\beta \rangle$ | ✓ | ✓ |
| SDW | $\mathcal{S}_{\alpha\beta\nu} = \langle \psi_\alpha^\dagger \sigma_\nu \vec{s}\, \psi_\beta \rangle$ | ✓ | ✓ |
| SC | $\Delta_{\alpha\nu} = \langle \psi_\alpha \sigma_\nu \psi_\alpha \rangle$ | ✓ | ✓ |
| Singlet exciton | $\Phi_{\alpha\pm}^C = \langle \psi_\alpha^\dagger \sigma_\pm \psi_\alpha \rangle$ | ✗ | ✓ |
| Triplet exciton | $\Phi_{\alpha\pm}^S = \langle \psi_\alpha^\dagger \sigma_\pm \vec{s}\, \psi_\alpha \rangle$ | ✗ | ✓ |
| PDW | $\mathcal{P}_{\alpha\beta\pm} = \langle \psi_\alpha^\dagger \sigma_\pm \psi_\beta^\dagger \rangle$ | ✗ | ✓ |

Notation: $\alpha$, $\beta$ index patch, $\sigma_\nu$ act on flavour, indexed by Latin characters $\nu = c$, $d$ with $\sigma_c = \frac{1}{2}(\sigma_0 + \sigma_z)$, $\sigma_d = \frac{1}{2}(\sigma_0 - \sigma_z)$, $\sigma_\pm = \frac{1}{2}(\sigma_x \pm i\sigma_y)$, and $\vec{s}$ is the vector of Pauli matrices acting on spin. The final two columns indicate if the given ordered state arises in the presence of a single nested vHS and/or TvHS

where $\beta_{g_{i,\nu}}$, $\beta_{h_i}$, $\beta_{j_i}$, $\beta_{l_i}$ are functions of all twenty couplings. Secondly, we compute the leading log$^2$ corrections to the order parameters, which generates the linear set of gap equations,

$$\frac{\partial}{\partial t}\mathcal{O}_i = \sum_j \mathcal{V}_{ij}(g, j, h, l)\mathcal{O}_j \tag{6}$$

where $\mathcal{O}_i = \{\mathcal{S}_{\alpha\beta\nu}, \mathcal{C}_{\alpha\beta\nu}, \Delta_{\alpha\nu}, \mathcal{P}_{\alpha\beta\pm}, \Phi_{\alpha\pm}^C, \Phi_{\alpha\pm}^S\}$. Diagonalising the gap equation matrix $\mathcal{V}_{ij}$ and integrating over the RG time $t$, one identifies the leading eigenvalue $\lambda_i(t)$, which diverges fastest with $t$. The associated eigenvector is the order parameter with the largest critical temperature $T_c = \Lambda e^{-1/(\nu_0 \lambda_i)^{1/2}}$, and is therefore the dominant order at $T \lesssim T_c$. Multiple orders of comparable $T_c$ may arise, in which case one must compute the Landau-Ginzburg free energy to ascertain whether such phases compete or coexist.

**Dominant instabilities**
A subset of the couplings diverge with increasing RG time $t \to \infty$. In this limit, the diverging couplings tend towards fixed constant ratios of each other referred to as *fixed rays*. The relative magnitudes of the couplings determine which ground state dominates. All possible choices of bare initial coupling values flow to one of these possible sets of ratios in the deep infrared, which therefore represent universal properties of the model. We now present the set of fixed rays possible in our TvHS patch model (for a derivation see the Supplementary Material). Despite the large number of interaction terms there turn out to be only a small set of fixed rays, shown in Fig. 2, which exhaustively characterise the possible ground states in the weak coupling regime. We summarise for three different cases:

1. Honeycomb systems with $\kappa = 1$ have three fixed rays: comprising chiral superconductivity $\Delta_d$, chiral $d$-wave excitons $\Phi_d^C$, and $s$-wave excitons $\Phi_s^C$.
2. Kagome systems with $\kappa = 1$ have seven fixed rays: comprising chiral superconductivity $\Delta_d$, chiral $d$-wave excitons $\Phi_d^{C/S}$, and $s$-wave excitons.
3. Kagome systems with $\kappa = 2$ have eight fixed rays: comprising chiral superconductivity $\Delta_d$, chiral $d$-wave excitons $\Phi_d^{C/S}$, and $s$-wave excitons $\Phi_s^{C/S}$.

Crucially, in all cases $d$-wave excitons emerge at a fixed trajectory, demonstrating the naturalness of excitonic order. As in the case of single vHS[45], we find that $d$-wave superconductivity is also a natural instability of the TvHS model.

For arbitrarily small initial couplings, the fixed rays are reached at long RG times, which corresponds to the deep infrared. However, the initial couplings could be sufficiently large that an instability occurs before the fixed ray is reached. In such a case it is appropriate to instead explicitly compute the flow from a specific set of initial conditions, and examine when an instability is reached. Given the significant magnitude of the bare values of the couplings in AV$_3$Sb$_5$

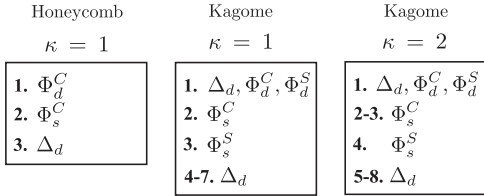

**Fig. 2 | RG Fixed rays.** RG fixed rays for the TvHS patch model in honeycomb and kagome systems. The parameter $\kappa$ measures particle-hole symmetry, c.f. discussion after Eq. (2); $\kappa = 2$ is appropriate to AV$_3$Sb$_5$[51].

(Table 1), we believe that such an analysis is more appropriate when comparing with experiment, and is presented below.

**Properties of the chiral excitonic condensate**
We focus attention on some key properties of the excitonic phases which appear. The two $d$-wave excitonic order parameter structures which appear are

$$\Phi_{\pm,a}^C = \Phi_a e^{\pm i\phi_a} \frac{1}{\sqrt{6}}(1, -2, 1),$$
$$\Phi_{\pm,b}^C = \Phi_b e^{\pm i\phi_b} \frac{1}{\sqrt{2}}(1, 0, -1) \tag{7}$$

Here, $\Phi_a$, $\Phi_b$ are real scalars, $\phi_a$, $\phi_b$ are distinct U(1) phases, and the row vectors enumerate patch indices, therefore encoding the spatial structure of the two order parameters. Continuing to the full Fermi surface, the spatial vectors schematically behave as $\sim \cos(2\theta_{\boldsymbol{k}})$, $\sin(2\theta_{\boldsymbol{k}})$, with $\theta_{\boldsymbol{k}}$ the momentum angle. Similar $d$-wave eigenvectors appear for the superconducting states $\Delta$, which are the two-flavour analogues of the superconducting states found in[45].

Near the critical temperature, the Landau-Ginzburg free energy is found to be

$$\mathcal{F}_\Phi = \mathcal{F}_0 + \left(\frac{1}{2\lambda_\Phi} - a_\Phi\right)(|\Phi_a|^2 + |\Phi_b|^2)$$
$$+ c_\Phi\left(\Phi_a^4 + \Phi_b^4 + \frac{4}{3}\Phi_a^2\Phi_b^2 + \frac{2}{3}\Phi_a^2\Phi_b^2\cos(2(\phi_a - \phi_b))\right) \tag{8}$$

where $\mathcal{F}_0$ is the free energy of the free fermions, with expansion coefficients $a_\Phi, c_\Phi > 0$. The free energy is minimised by coexisting order parameters, with $\Phi_a = \Phi_b = \Phi_0$ and $\phi_a - \phi_b = \pi/2$ (mod $\pi$). The coexisting states form a single order parameter of the form $\Phi = \Phi_0 e^{\pm i\theta_\alpha}$, $\theta_\alpha = \{a, b, c\}$. Continuing around the Fermi surface, the combined order parameter becomes $\Phi \sim \Phi_{0,\boldsymbol{k}}(\cos(2\theta_{\boldsymbol{k}}) \pm i\sin(2\theta_{\boldsymbol{k}}))$, which is a chiral $d \pm id$ order. The chirality $\pm$ is spontaneously selected by the ground state, which therefore breaks time-reversal symmetry.

In addition to broken TRS, the chiral order parameter winds twice along the Fermi surface and vanishes away from it, thereby exhibiting a non-trivial topology with Chern number $|C| = 2$. In order to illustrate this, we diagonalise a mean-field Hamiltonian (Methods III) defined on a lattice—we consider a two-orbital kagome lattice model in an infinite ribbon geometry with zigzag edges. The 1D dispersion of the ribbon is plotted in Fig. 3 for the $d + id$ phase, which exhibits two chiral edge modes, with the left/right-movers propagating along the top/bottom of the ribbon. Full details of the lattice model are provided in the Supplementary Material. The non-trivial topological invariant implies a quantised anomalous Hall conductivity $\sigma_{xy} = Ce^2/(2\pi)$, which is carried by two chiral edge modes. We note that this value of $\sigma_{xy} = Ce^2/(2\pi)$ accurately accounts for the intrinsic contribution to the anomalous Hall effect seen in AV$_3$Sb$_5$[17].

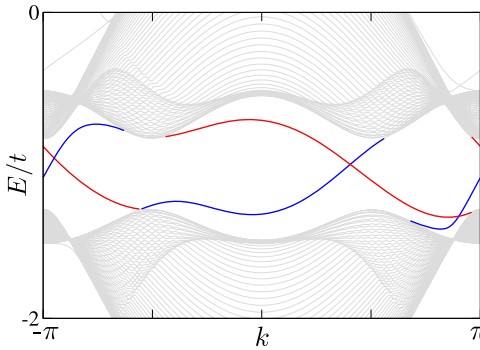

**Fig. 3 | Edge modes.** The 1D dispersion of a $d + id$ excitonic insulator in a two-orbital kagome system, for an infinite ribbon of width 120 unit cells. The edge states propagating along the top/bottom of the ribbon are plotted in red/blue.

## Coexistence of chiral excitons and charge order

The RG procedure determines which phases are dominant, but is not sufficient to determine the actual ground state when two phases have comparable $T_c$. Motivated by experiments on AV₃Sb₅, we examine the ground state when CDW and chiral excitons are proximate instabilities. To carry out the analysis, we consider the Landau-Ginzburg free energy for chiral excitons and CDW, written as $\mathcal{F} = \mathcal{F}_\Phi + \mathcal{F}_C + \mathcal{F}_{\Phi,C}$, with $\mathcal{F}_\Phi$ from Eq. (8) and

$$
\begin{aligned}
\mathcal{F}_C = & \sum_{\alpha \neq \beta; \nu = c, d} \left\{ (a_\nu \delta_{\nu\nu'} + \tfrac{1}{2}\mathcal{V}_{\nu\nu'}^{-1}) C_{\alpha\beta\nu} C_{\alpha\beta\nu'}^* + \tfrac{1}{2} c_{1\nu} |C_{\alpha\beta\nu}|^4 \right\} \\
& + \sum_\nu b_\nu \{ C_{12\nu} C_{23\nu} C_{31\nu} + \text{c.c.} \} + \sum_{\alpha \neq \beta; \nu \neq \nu'} c_{2\nu} |C_{\alpha\beta\nu}|^2 |C_{\alpha\beta\nu'}|^2 \\
\mathcal{F}_{\Phi,C} = & \sum_{\alpha \neq \beta; \nu \neq \nu'} c_{3\nu} \{ C_{\alpha\beta\nu} C_{\alpha\beta\nu'}^* \Phi_\alpha \Phi_\beta^* + \text{c.c.} \} \\
& + \sum_{\alpha \neq \beta, \nu} c_{4\nu} |C_{\alpha\beta\nu}|^2 (|\Phi_\alpha|^2 + |\Phi_\beta|^2).
\end{aligned} \tag{9}
$$

The expansion coefficients $a_\nu, b_\nu, c_{i\nu}$ depend on temperature and are computed in the Supplementary Material. Here $\mathcal{V}_{\nu\nu'}$ is the CDW gap equation matrix (6). Henceforth, we denote the largest eigenvalue of $\mathcal{V}_{\nu\nu'}$ by $\lambda_{\text{CDW}}$.

In the free energy (9), there are six complex numbers, $C_{\alpha\beta\nu}$, describing CDW order. Physically the $C_{\alpha\beta\nu}$ correspond to the magnitude of the order for the three distinct vectors $\boldsymbol{Q}_{\alpha\beta} \in \{\boldsymbol{Q}_{12}, \boldsymbol{Q}_{23}, \boldsymbol{Q}_{31}\}$, on the two distinct Fermi surface flavours. From the gap equation (6) we find that the leading CDW order has $C_{\alpha\beta c} = \rho C_{\alpha\beta d}$, where $\rho$ is a real number. In particular, $|\rho| = 1$ in the particle-hole symmetric limit of $\kappa = 1$. Moreover, the gap equation distinguishes real charge density order (rCDW) whereby $C_{\alpha\beta\nu}^* = C_{\beta\alpha\nu}$ and purely imaginary order (iCDW) whereby $C_{\alpha\beta\nu}^* = -C_{\beta\alpha\nu}$. We separately considered parameter regimes in which rCDW and iCDW were the leading CDW phase.

We turn now to the phase diagram predicted by (9). To construct the phase diagrams of Fig. 4(a)i and 4(b)i we allow the eigenvalues $\lambda_{\text{CDW}}$ and $\lambda_\Phi$ to be free variables. To illustrate the property of coexistence and our phenomenological proposal for these AV₃Sb₅, we take a realistic range of coupling eigenvalues, consistent with DFT calculations[51], and fix $T = 80$ K (which enters via the coefficients $a_\nu, b_\nu, c_{i\nu}$). We find three distinct phases: (i) chiral excitons, (ii) CDW, and (iii) coexistence of chiral excitons and CDW. In the rCDW/iCDW phases, the 3$\boldsymbol{Q}$ state is favoured, i.e., CDW order is nonzero for all three nesting vectors $\boldsymbol{Q}_{\alpha\beta}$, and corresponds to $C_{12\nu} = C_{23\nu} = C_{31\nu} \neq 0$. In the excitonic phases, the chiral (TRS breaking) $d + id$ state is favoured. In the region of coexistence, chiral excitons and the 3$\boldsymbol{Q}$ CDW are favoured. We point out that $d$-wave excitons coupled to a nematic CDW (e.g., $C_{12\nu} > C_{23\nu} = C_{31\nu}$) was observed as a local minima, but did not appear as the global minimum over the parameter range searched.

Experiment indicates that TRS breaking and CDW coexist in AV₃Sb₅, and set in at $T^* \approx 100$ K. We propose that the coexistence phase demonstrated by our Landau-Ginzburg analysis provides a phenomenological explanation of the physics of kagome metals at $T \lesssim T^*$.

### Truncated RG flow and phase diagram for AV₃Sb₅

To complement the analysis leading to Fig. 4(a)i and 4(b)i, we now directly compute the eigenvalues $\lambda_{\text{CDW}}$ and $\lambda_\Phi$ from the RG procedure. Unlike for the fixed ray analysis, here we must provide initial conditions for the RG flow. Once initialised, we perform the RG flow down from a UV scale of $\Lambda \approx 0.5$ eV to an infrared scale set by $T$. We use the resulting renormalised couplings as input to the free energy, minimising to obtain the resulting ground state. This procedure generates the phase diagrams of Fig. 4(a)ii and (b)ii.

We discuss now the choice of initial couplings that lead to Fig. 4(a) ii and (b)ii. Given that several of the couplings in Table 1 vanish, we allow for the situation where these couplings take negative values. By inspection of the gap equation (6), we see that an initial value of $g_{1,i} < 0$ promotes CDW (this was first noted in[52] for the problem of a single vHS), while $h_1 < 0$ promotes chiral excitons. To this end, we first allow for both $g_{1,c}, h_1 < 0$, and subsequently arrive at the phase diagram of Fig. 4(a)ii. In addition, we allow for $g_{3,d} = 0$, and arrive at Fig. 4(b)ii. The phase diagram is qualitatively the same for $g_{3,d} < 0$. Next we mention that the magnitudes and ratios have been estimated from ab initio calculations (Table 1). A more accurate treatment would account for the renormalisation of the couplings in going from lattice to the patch UV cutoff $\Lambda$. We have not included these effects in our analysis. We stress that our use of the values in Table 1 is to illustrate that there exist physically reasonable bare couplings, which produce the desired phenomenology.

## Discussion

We introduced and analysed a minimal model to describe interacting fermions near a twofold van Hove singularity (TvHS)—two opposite concavity vHS near the Fermi level. We found the opposite concavities of the two vHS crucially affect the possible many-body instabilities, relative to the single vHS case. In particular, excitonic order contends as a possible instability and generically results in a chiral $d$-wave excitonic phase in hexagonal systems such as honeycomb and kagome lattices. We contrast our scenario with topological excitonic states, which have been previously explored theoretically[53–56]; in our case, the topology of the $d + id$ condensate is not inherited from the Berry curvature at the $K$-points or from spin–orbit coupling, but appears at the $M$-point intrinsically due to interaction driven, spontaneous time-reversal symmetry breaking. These findings suggest a new class of candidate materials for topological excitonic ground states.

TvHS were recently seen experimentally in AV₃Sb₅[39]. We now discuss key features of experiment and the extent to which the TvHS minimal model explains them: First, signatures of time-reversal symmetry breaking, including a significant anomalous Hall effect, are observed at temperatures near to $T^*$ despite the lack of magnetic ordering[19,41]. The presence of chiral excitonic order would offer an appealing interpretation of the broken time-reversal symmetry and anomalous Hall effect. Second, experiments also report the breaking of threefold rotational symmetry and onset of nematic order around $T_c \lesssim 50$ K. Coupling between excitons and CDW naturally results in a phase consisting of nodal $d$-wave excitons and a nematic CDW, however, our analysis of the free energy found this phase was only ever a local minimum in our model. Coupling to phonons may promote this phase to the dominant ground state, and we leave further examination of this scenario to future work. Third, superconductivity emerges generically as an instability of the TvHS minimal model. However, superconductivity is seen at a much lower temperature scale ($T_c \approx 3.5$ K)[17,18,20–24] than CDW ($T^* \approx 100$ K). At these temperatures the

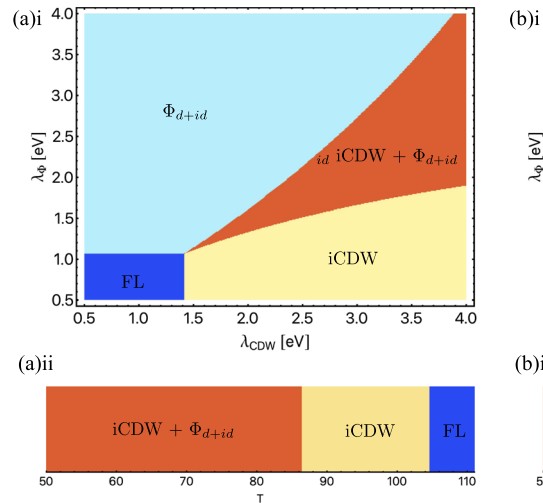

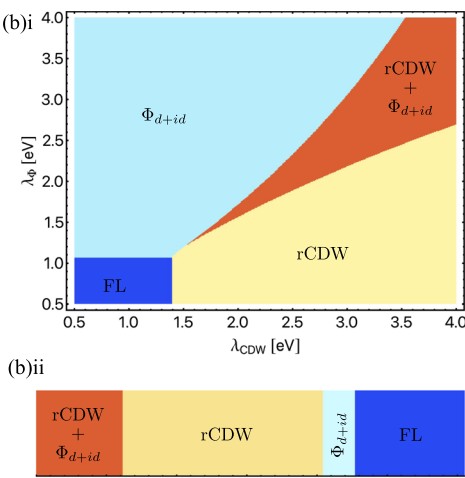

**Fig. 4 | Phase diagrams.** Everywhere we have taken $\kappa = 2$; i/rCDW + $\Phi_{d+id}$ represents coexistence, and FL the Fermi liquid metallic state. **a**(i) The iCDW + $\Phi_{d+id}$ and **b**(i) the rCDW + $\Phi_{d+id}$ phase diagrams, using the eigenvalues $\lambda_\Phi$, $\lambda_{CDW}$ as free parameters. In **a**(ii) and **b**(ii) the eigenvalues $\lambda_\Phi$, $\lambda_{CDW}$ are computed explicitly by solution of the RG equations, resulting in a phase diagram as a function of $T$. The two different phase diagrams result from taking **a**(i) $g_{1,c} < 0$, $h_1 < 0$, and **b**(ii) $g_{1,c} < 0$, $h_1 < 0$ with $g_{3,d} = 0$; precise values of the initial couplings are given in the Supplementary Material.

correct starting point for a description of superconductivity requires incorporating the CDW into the ground state.

Time-reversal symmetry (TRS) breaking and the anomalous Hall conductivity has also been proposed to arise due to a complex CDW state[42,43,52]. Our analysis shows that it is possible for CDW and chiral excitons to coexist, but a key difference between these two states is that chiral excitons break TRS but not translational symmetry, allowing experiment to disentangle the two. To this end, it has been shown that the application of strain and pressure can selectively move the two opposite concavity Fermi surfaces closer or further away from the Fermi level[57]. Moving the Fermi level away from one of the vHS creates a mismatch in the Fermi surface areas, which disfavours the excitonic phase, whereas moving the Fermi level away from the two vHS in a way that keeps the Fermi surface areas roughly equal disfavours charge order, while retaining the tendency to excitonic order. Experimental probes of TRS breaking could be applied in the presence of strain and pressure to disentangle the two phases. Additionally, we suggest that the existence of an exciton condensate should lead to Andreev-like interband tunnelling, an effect which has been used to probe excitonic order in bilayer systems[58].

Beyond the vanadium metals $AV_3Sb_5$, another possible scenario for topological excitonic condensation is to engineer TvHS in van der Waals heterostructures made from materials with hexagonal symmetry such as graphene or transition metal dichalcogenides (TMDCs)[59,60]. In moiré systems, the reduced bandwidth of the bands near charge neutrality brings the opposite concavity vHS of the valence and conduction bands closer in energy, so that an bias field could feasibly result in a TvHS. The valence and conduction bands can be further aligned through spin splitting the bands via a TMDC layer[61,62], the effect of which can be tuned via twist angle[63]. Finally, valley polarisation is observed in twisted layered systems[64–66], which could also be exploited to align the valence and conduction bands, e.g., through methods discussed in refs. [67,68]. In the context of layered van der Waals materials, a possible experimental probe would be the enhanced tunnelling between layers induced by excitons, e.g., refs. [69,70].

## Methods
### Leading instabilities
Our discussion of the leading ordered states follows from the solution of the gap equations for the order parameter vertices

$\mathcal{O}_i = \{\mathcal{S}_{\alpha i}, \mathcal{C}_{\alpha i}, \Delta_{\alpha i}, \mathcal{P}_{\alpha\pm}, \Phi^C_{\alpha\pm}, \Phi^S_{\alpha\pm}\}$. We find the mean-field gap equations to be

$$\frac{\partial}{\partial t}\Phi^C_{\alpha+} = d_4 \sum_{\beta\neq\alpha}\left\{(h_4 - 2l_4)\Phi^C_{\alpha+} - j_4\Phi^C_{\alpha-} + (h_1 - 2l_2)\Phi^C_{\beta+} + (j_1 - 2j_2)\Phi^C_{\beta-}\right\}$$

$$\frac{\partial}{\partial t}\Phi^S_{\alpha+} = d_4 \sum_{\beta\neq\alpha}\left\{h_4\Phi^S_{\alpha+} + j_4\Phi^S_{\alpha-} + h_1\Phi^S_{\beta+} + j_1\Phi^S_{\beta-}\right\}$$

$$\frac{\partial}{\partial t}\mathcal{P}_{\alpha+} = -d_1\{h_2\mathcal{P}_{\alpha+} + h_1\mathcal{P}_{\alpha-} + l_1\mathcal{P}_{\alpha+} + l_2\mathcal{P}_{\bar\alpha-}\}$$

$$\frac{\partial}{\partial t}\mathcal{C}_{\alpha,\nu} = d_{2\nu}(g_{2,c} - 2g_{1,c})\mathcal{C}_{\alpha,c} - d_{2\nu}g_{3,c}\mathcal{C}_{\bar\alpha,c} + d_{2\nu}(l_2 - 2h_1)\mathcal{C}_{\alpha,d} + d_{2\nu}(l_3 - 2h_3)\mathcal{C}_{\bar\alpha,d}$$

$$\frac{\partial}{\partial t}\mathcal{S}_{\alpha,\nu} = d_{2\nu}g_{2,c}\mathcal{S}_{\alpha,c} + d_{2\nu}g_{3,c}\mathcal{S}_{\bar\alpha,c} + d_{2\nu}l_2\mathcal{S}_{\alpha,d} + d_{2\nu}l_3\mathcal{S}_{\bar\alpha,d}$$

$$\frac{\partial}{\partial t}\Delta_{\alpha,\nu} = -\sum_{\beta\neq\alpha}\left\{d_{0\nu}g_{4,c}\Delta_{\alpha,c} + d_{0\nu}g_{3,c}\Delta_{\beta,c} + d_{0\nu}j_4\Delta_{\alpha,d} + d_{0\nu}j_3\Delta_{\beta,d}\right\}$$

(10)

with indices as defined previously: $c$, $d$, $\pm$ referring to flavour, $\alpha$ to patch, and $\bar\alpha$ denoting the patch connected to $\alpha$ by a nesting vector. To make the equations compact, we have introduced $\nu = \{c, d\}$ with $\bar\nu = \{d, c\}$. The $d$-factors are nesting coefficients that characterise the relative strength of the particle-particle and particle-hole divergences, and are defined in the Supplementary Material—we have used notation so that $d_{0c} = 1$, $d_{0d} = d_0$, $d_{2c} = d_2$, $d_{2d} = d_3$. The couplings entering the gap equations are understood to inherit scale-dependence from the RG equations for the couplings (5). The eigenvectors for this linear system of gap equations give the possible order parameter structures, and those with the largest eigenvalue are the leading instabilities. The set of Feynman diagrams which generate these flow equations are given in the Supplementary Material.

### Landau-Ginzburg analysis
The susceptibility gap equations (10) are insufficient to determine whether order parameters compete or can form a ground state in which multiple orders coexist. Given a set of degenerate or nearly degenerate solutions to the gap equations, we determine which combination of these solutions is the favoured ground state by calculating the Landau-Ginzburg free energy. We employ the mean-field decomposition of the fermions coupled to a combination of order parameter matrices, and integrate out the fermionic degrees of

freedom, arriving at the free energy

$$\mathcal{F} = \frac{1}{2\lambda_\Phi} \sum_i |\Phi_i|^2 + \frac{1}{2} \sum_{\alpha \neq \beta; \nu = c,d} \mathcal{V}_{\nu\nu'}^{-1} C_{\alpha\beta\nu} C_{\alpha\beta\nu'}^* - \text{Tr} \log \mathcal{G}^{-1}. \tag{11}$$

Here the full Green's function

$$\mathcal{G}^{-1}(i\omega_n, \boldsymbol{q}) = \mathcal{G}_0^{-1}(i\omega_n, \boldsymbol{q}) + M, \tag{12}$$

comprises the order parameter matrix $M = M_\Phi + M_C$,

$$M_\Phi = \begin{pmatrix} \Phi_1 & 0 & 0 \\ 0 & \Phi_2 & 0 \\ 0 & 0 & \Phi_3 \end{pmatrix} \otimes \sigma_+ + \begin{pmatrix} \Phi_1^* & 0 & 0 \\ 0 & \Phi_2^* & 0 \\ 0 & 0 & \Phi_3^* \end{pmatrix} \otimes \sigma_- \tag{13}$$

$$M_C = \begin{pmatrix} 0 & C_{12c} & C_{31c}^* \\ C_{12c}^* & 0 & C_{23c} \\ C_{31c} & C_{23c}^* & 0 \end{pmatrix} \otimes \sigma_c$$
$$+ \begin{pmatrix} 0 & C_{12d} & C_{31d}^* \\ C_{12d}^* & 0 & C_{23d} \\ C_{31d} & C_{23d}^* & 0 \end{pmatrix} \otimes \sigma_d \tag{14}$$

and the bare Green's function

$$\mathcal{G}_0^{-1}(i\omega_n, \boldsymbol{q}) = \begin{pmatrix} i\omega_n - \varepsilon_1(\boldsymbol{q}) & 0 & 0 \\ 0 & i\omega_n - \varepsilon_2(\boldsymbol{q}) & 0 \\ 0 & 0 & i\omega_n - \varepsilon_3(\boldsymbol{q}) \end{pmatrix} \otimes \sigma_c$$
$$+ \begin{pmatrix} i\omega_n + \varepsilon_1(\boldsymbol{q}) & 0 & 0 \\ 0 & i\omega_n + \varepsilon_2(\boldsymbol{q}) & 0 \\ 0 & 0 & i\omega_n + \varepsilon_3(\boldsymbol{q}) \end{pmatrix} \otimes \sigma_d. \tag{15}$$

The dispersion at each patch is $\varepsilon_1(\boldsymbol{q}) = \frac{1}{2} q_x(q_x + \sqrt{3}q_y)$, $\varepsilon_2(\boldsymbol{q}) = \frac{1}{4}(-q_x^2 + 3q_y^2)$ and $\varepsilon_3(\boldsymbol{q}) = \frac{1}{2}q_x(q_x + \sqrt{3}q_y)$. For the two degenerate $d$-wave excitons, parameterised by $\Phi_a$ and $\Phi_b$, we have

$$\Phi_1 = -\frac{1}{\sqrt{2}}\Phi_a - \frac{1}{\sqrt{6}}\Phi_b,$$
$$\Phi_2 = \sqrt{\frac{2}{3}}\Phi_b, \tag{16}$$
$$\Phi_3 = \frac{1}{\sqrt{2}}\Phi_a - \frac{1}{\sqrt{6}}\Phi_b.$$

Rewriting

$$\text{Tr} \log \mathcal{G}^{-1} = -\mathcal{F}_0 + \text{Tr} \log(1 + \mathcal{G}_0 M) \tag{17}$$

where $\mathcal{F}_0$ is the free energy of a free Fermi gas, and using the expansion

$$\text{Tr} \log(1 + \mathcal{G}_0 M) = \sum_{n=0}^{\infty} \frac{(-1)^n}{n} \text{Tr}(\mathcal{G}_0 M)^n \tag{18}$$

we evaluate the trace of the first four terms in the expansion, resulting in the free energy stated in the main text. Determining whether the minimum of the free energy can include coexisting $C_{\alpha\beta\nu}$ and $\Phi_a$, $\Phi_b$ requires knowledge of the coefficients in this expansion; their calculation is detailed in the Supplementary Material.

## Edge states

To demonstrate the presence of edge states in the excitonic phase, we employ a simplified model for numerical diagonalisation, describing a kagome lattice with two-orbital states $\nu = \pm$,

$$H = -\sum_{\langle \boldsymbol{r}, \boldsymbol{r}' \rangle, \nu} t_\nu c_{\boldsymbol{r}',\nu}^\dagger c_{\boldsymbol{r},\nu} + \sum_{\boldsymbol{r}} \gamma_0 c_{\boldsymbol{r},1}^\dagger c_{\boldsymbol{r},1}$$
$$+ \sum_{\langle \boldsymbol{r}', \boldsymbol{r} \rangle} \Delta(\boldsymbol{r}', \boldsymbol{r}) c_{\boldsymbol{r}',\nu}^\dagger c_{\boldsymbol{r},\nu} + \text{h.c.} \tag{19}$$

in which only coupling between nearest neighbours is taken into account. We choose the excitonic pairing function $\Delta(\boldsymbol{r}, \boldsymbol{r}')$ so that the lattice theory possesses an equivalent continuum limit to our field theory description of the three patches surrounding the $M$ points. The gap function is

$$\Delta(\boldsymbol{r}', \boldsymbol{r} \in \sigma) = \frac{1}{\sqrt{6}} \Delta_0 e^{i(\theta_{\boldsymbol{r}'-\boldsymbol{r}} - (\ell+1)\varphi_\sigma)},$$
$$(\varphi_A, \varphi_B, \varphi_C) = \left(0, \frac{2\pi}{3}, \frac{4\pi}{3}\right), \tag{20}$$

with $\ell = \pm 2$ equal to the phase winding of the excitonic order around the Fermi surface. The results for $\ell = -2$ are plotted in Fig. 3 in the main text with $\gamma_0 = 2t$, $\Delta_0 = 0.5t$, for a ribbon geometry with 60 unit cells.

## Data availability
The data produced in this study are available upon reasonable request.

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

## Acknowledgements

We thank Brenden Ortiz, Michael Denner, Mingu Kang, Colin Nancarrow and Dmitry Efimkin for discussions and comments. H.S. and O.P.S. acknowledge funding from ARC Centre of Excellence FLEET. T.L. acknowledges support from the Deutsche Forschungsgemeinschaft.

## Author contributions

H.D.S. and J.I. conceived of the project idea, performed the RG analysis and calculated the Landau-Ginzburg free energy. T.L. performed edge state calculations. O.P.S. critically assessed the project idea and calculations. All authors discussed the details and contributed to writing.

## Competing interests

The authors declare no competing interests.
