## [Peer Review File · Nature Communications]

Chiral excitonic order from twofold van Hove singularities in kagome metalsREVIEWER COMMENTS

Reviewer #1 (Remarks to the Author):

The manuscript by Scammell et al employs the perturbative renormalization group technique to study the many-body electronic instabilities in Kagome-honeycomb lattice heterostructure with twofold van Hove singularities. Interestingly, they identified a topological d-wave excitonic instability, the ordering of which can give rise to observable effects such as anomalous Hall effect (AHE). The authors claim that their findings may shed light on understanding the puzzling features in recently extensively studied kagome metal AV3Sb5 (A=K, Cs, Rb), as well as be realized in bilayer materials.

Both the two proposals for material realizations are quite topical, in particular, the discovery of AV3Sb5 (A=K, Cs, Rb) has generated a lot of research interests, with a few puzzling questions to be understood. For example, what is the nature of translation symmetry breaking order near 100K, how it is related to the anomalous Hall effect, rotational symmetry breaking observed below about 100K. There have been proposals on a time-reversal breaking charge order, which can partially explain some experiments, such as the μ SR, Kerr rotation and (arguably) AHE.

The authors provide a different perspective based on their finding of the d+id excitonic order due to twofold van Hove singularities. I think this is a well motivated toy model beyond the previously considered single band model. However, I have questions on how the new physics obtained here, i.e. d+id excitonic order, provide new insights on understanding the puzzling experiments in AV3Sb5 compounds. My concerns are the following: First, there have been many experimental evidence for a translation symmetry breaking charge order near 100K, but it is not present in the phase diagram obtained by the authors, as the d+id excitonic order carries zero momentum. Second, as far as I understand from the manuscript, the only new signature of the excitonic order that is not present with only charge order (time reversal even or odd) is the claim "but DFT calculations and magnetotransport indicate that the roughly hexagonal Fermi contours hosting the twofold van Hove singularities remain in the reconstructed band structure, which mainly undergoes changes near the K points [50]" (quoted from the IV Discussion section of the manuscript). However, from other studies, e.g. Ref. 1, the twofold van Hove band structure is reconstructed at M points. For this reason, to provide insights on the AV3Sb5, I think the authors should make a more careful comparison with the charge ordering scenario, and explain to what extent, their findings can fit into the observation of translation symmetry breaking charge order. On the other hand, given the puzzles to reconcile various experiments in AV3Sb5, I agree that different ideas are very valuable, and the excitonic order, if realized, would be very cool. One theoretical possibility that I can think of is to generalize the RG framework developed by the authors, and look for a possible fixed point with charge order as the leading instability, and the excitonic order as a subleading instability. To generate charge order, the authors may consider introducing some negative bare interactions, as suggested in Ref. 2.

Because I think shedding insights into currently extensively studied systems, i.e. AV3Sb5, is an important factor to consider this manuscript publishing in a high profile journal like Nature Communications, I cannot recommend the current version of the draft to publish in Nature Communications unless more explanations and summaries on the new insights on AV3Sb5 can be provided.

Besides the major problem discussed above, below are some detailed questions and comments on the manuscript.

1. In Eq. 5, the authors presented the pp and ph polarization bubble for intra and inter layer states with leading double logarithm. I have questions about the coefficients, are these for different channels really the same ($=\nu_0/4$)? At least, I think the Π_{pp}^{ii} and Π_{ph}^{ii} are different by a factor 1/2 (see Appendix B in Ref. 3) . Naively, I would think d1 and d3 are different as well.

2. I think the manuscript should have more discussions on how the phase diagram depends on various parameters, such as the d_a coefficients introduced in Eq (5), the bare couplings. As far as

I can tell, the phase diagram is not obtained from any fixed "point", but from the running of coupling coefficients. I am worried that the phase diagram can be sensitive to bare parameters as it is not from RG fixed "point". In particular, what is the true fixed point for the phase diagram shown in Fig. 2?

3. The manuscript focus on log divergent van Hove singularities for both m and p bands. To apply to AV3Sb5 with greater accuracy, one of the band should be a quartic-saddle-point as the authors acknowledged. It would be very nice if there can be some discussions on how this could modify the phase diagram, even at a qualitative level.

References:

1. M. Kang et al, Nature Physics, 18, 301-308 (2022)
2. T. Park et al, Phys. Rev. B 104, 035142 (2021)
3. Y. Lin et al, Phys. Rev. B 100, 085136 (2019)

Reviewer #2 (Remarks to the Author):

The manuscript presents a model study of a system of interaction fermions on a hexagonal lattice geometry featuring two active electronic bands near van-Hove filling where the two bands exhibit opposite concavity – a configuration which has been coined "two-fold van-Hove singularity". This specific configuration facilitates the formation of a d-wave excitonic condensate, which the authors argue to be a topologically non-trivial state of matter, but also unconventional superconductivity. The model construction is inspired by recent experiments on kagome metals, which found evidence for unconventional superconductivity and time-reversal symmetry breaking.

To investigate the correlated states of their model and their competition, the authors employ a standard parquet RG and they also carry out Landau-Ginzburg analysis for the interesting excitonic condensate state, which is argued to feature a quantum anomalous Hall conductance. They further discuss the response of the system to variations of the filling and effective interaction parameters.

The manuscript is generally well written, and well-structured and it is overall a pleasant read. The considered problem is well introduced and properly motivated. The work is technically involved, mainly because of the large number of couplings in the RG approach, but at the same time it is methodically quite straightforward employing the well-established parquet RG. The concomitant technicalities of the method are introduced in a self-contained way with ample details postponed to the appendices. References and the concluding discussion are appropriate. Also, I find the essential results of the manuscript and the general scenario very interesting and I believe that the work may provide significant impact.

I have, however, a major reservation when it comes to recommending the work for publication in Nature Communications. In my opinion, the presentation of the work is rather technical, i.e. I would consider the general style to be exceedingly appropriate for example for the APS Physical Review journals. For Nature Communications, on the other hand, I would expect the essential physical results to be presented in a manner that is more accessible to its broad readership. In particular, I think that the discussion of the interesting topological d-wave excitonic condensate state and its underlying physical mechanism deserves to be more clearly exposed in the manuscript, which I believe is possible without resorting to an overly technical discussion. So, my central point of criticism is that the manuscript, if considered for Nature Communications, requires a major revision concerning the presentation of the results according to the principle indicated above.

I append two additional comments/questions that should be addressed in a revised version of the manuscript, independent of the publication venue:

- (1) One important shortcoming of the presented approach is that it does not allow for a real

material-specific prediction about the leading Fermi-surface instability, because it only resolves the very low-energy excitations around the saddle-points of the energy dispersion. Therefore, no realistic Coulomb interactions can be taken into account. Instead, the authors have to choose more or less arbitrary initial values for their couplings g, h, j, l . Nevertheless, it is known that the strength and range of the Coulomb interactions can have a profound impact on the ground state. The parquet RG reflects this fact partly by featuring several RG fixed trajectories which are realized for different initial combinations of the couplings g, h, j, l . The authors have only considered a very small subspace of the corresponding parameter space, potentially disregarding a very large amount of interesting correlated states. I think that it would be interesting to provide a more exhaustive account for the possible RG fixed trajectories and their concomitant correlated states.

(2) The authors mention that the ARPES data on the vanadium metals seem to feature a higher-order van-Hove singularity (due to the quartic saddle point) at least in one of the bands. This is a very interesting remark, but at the same time it calls the applicability of the presented results on the specific material into questions. As has been studied elsewhere, the presence of higher-order van-Hove singularities substantially changes the hierarchy of leading contributions to the RG evolution equations and therefore also alters the leading Fermi-surface instabilities. I think that the authors need to provide an argument on why this fact does not fundamentally change their results or they need to expose more clearly that they are not obviously applicable to the vanadium metals.

Reviewer #3 (Remarks to the Author):

The manuscript "Chiral excitonic order, quantum anomalous Hall effect, and superconductivity from twofold van Hove singularities in kagome metals" is a numerical study of the leading electronic instabilities in a model inspired by the recently discovered class of kagome metals. The authors use perturbative renormalization group analysis and focus on the effect of the multiple van Hove singularities present in the multi-orbital kagome low-energy model, put forward previously for these compounds.

The topic is timely and the results reported are novel. The finding of an excitonic d-wave order parameter and the connection to quantum anomalous Hall response have the potential to become relevant in the fast growing field of kagome metals.

However, before supporting publication on Nature Communications, I would like the following points to be more elaborated on by the authors:

1) Choice of interaction: even if the model on which the authors focus is highly idealized, the importance of this study strongly relies on its connection to the realistic compounds. For this reason, the choice of the multi-flavor interaction in Eq. 3 must -- at least qualitatively -- be motivated by a first-principle calculation. Recently the first cRPA calculation have started to appear in the literature [see e.g. <https://doi.org/10.48550/arXiv.2203.05038>] so the authors can connect to the values reported there. In this respect, I suggest adding a discussion on the choice of the coupling strength in the flavor basis (not in the band basis) as well as a check of the robustness of the main outcome discussed, against changes of these coupling strengths in the ballpark of the cRPA values.

2) The authors stress the opposite concavities of the van Hove singularities but do not consider that the "pure" and "mixed" ones differ also for their spatial support. As discussed in Ref. 9, 8 and 11, and shown in Ref. 53 (see Fig. 3e-g of Ref. 53) the decisive difference between them is their complementary sublattice localization (from which their names). The Bloch wave function of the pure van Hove localizes only on one sublattice (more precisely, the charge profile at each of the three M points has a maximum on one of the three sublattice of the kagome unit cell) while the Bloch wave function of the mixed van Hove localizes on the remaining two sublattices, respectively. This important distinction, first discussed in Refs 9, 8, and 11 and recently illustrated

and supported by measurements in Ref. 53 has to be explained in the present manuscript, otherwise the only distinction seems to be the hole- or electron-like character of these bands. An extremely interesting thing would be to explicitly analyze the effect of this spatial support on the different types of instabilities discussed here, given the fact that pure and mixed van Hoves are believed to react different to local and non-local Coulomb terms. One further relevant point is to analyze the relative importance of the concavity compared to the spatial support of these bands for the d-wave excitonic order. The manuscript would profit strongly if a discussion on these points, based on the numerical analysis of the authors, could be added. Moreover, at the end of section II B, the above sublattice support should be introduced as important property of the kagome van Hove bands, in addition to their quadratic/quartic momentum dispersion.

minor points:

-in Fig. 1b it could be more explicitly indicated that the upper band comes from the kagome because, given the restricted y -range, one does not see the flat band. Of course, looking at the sign of the parameter γ_0 one can get this information but it is not immediately direct for a reader.

-the phase diagrams in Fig. 2 seem really sketched by hand rather than been drawn on the basis of some results. The authors write "we arrive the the phase diagram Figure 2." (BTW, watch out the typo) but I guess some numbers at least could be added to give these plots a bit of quantitative look

Resubmission of NCOMMS-22-03563-T: “Chiral excitonic order, quantum anomalous Hall effect, and superconductivity from twofold van Hove singularities in kagome metals”

List of key changes:

1. **Presentation.** The manuscript has undergone major revisions, with a focus on emphasising key qualitative features and elucidating the physical mechanisms underpinning chiral excitons and CDW in a more accessible format. We have kept a high-level discussion of the methods/techniques in the main text, and relegated explicit expressions for the interactions and gap equations to the supplement, with some key details in a methods section. The main text now predominantly focuses on the key concepts.
2. **Charge order in TvHS.** Excitons are proven to coexist with, rather than compete against, charge density wave order. Charge order is shown to be nearly degenerate with excitonic order for experimentally relevant parameters, resulting in a phase diagram featuring both phases. These updated results make direct connection to experiment.
3. **Discussion of sublattice support.** We extend our theoretical modelling to account for different bandwidths and orbital content on each Fermi surface. This allows us to separately investigate honeycomb and kagome systems, i.e. with and without the effects of sublattice interference.
4. **Initial conditions from ab initio calculations.** To make closer contact with experiments on AV_3Sb_5 , we take interaction constants from ab initio calculations, and use them to estimate the bare couplings in our model. From these initial conditions, our model produces a quantitative phase diagram [Fig. 4].
5. **More exhaustive RG treatment.** We present an exhaustive list of the RG fixed rays in both honeycomb and kagome systems. This gives the universal set of weak coupling instabilities. Crucially, in all cases chiral excitonic order appears. This analysis complements our results of Fig. 4, which used ab initio parameters as input – and is therefore non-universal.
6. **Influence of higher-order vHS.** We add discussion of the influence of quartic corrections to the saddle point. We find that for AV_3Sb_5 , the quartic corrections can be neglected for the energy scales probed in our analysis.
7. **Title change:** “Chiral excitonic order from twofold van Hove singularities in kagome metals”.

Point-by-point response to referee #1:

Referee: *The manuscript by Scammell et al employs the perturbative renormalization group technique to study the many-body electronic instabilities in Kagome-honeycomb lattice heterostructure with twofold van Hove singularities. Interestingly, they identified a topological d-wave excitonic instability, the ordering of which can give rise to observable effects such as anomalous Hall effect (AHE). The authors claim that their findings may shed light on understanding the puzzling features in recently extensively studied kagome metal AV_3Sb_5 ($A=K, Cs, Rb$), as well as be realized in bilayer materials.*

Both the two proposals for material realizations are quite topical, in particular, the discovery of AV_3Sb_5 ($A=K, Cs, Rb$) has generated a lot of research interests, with a few puzzling questions to be understood. For example, what is the nature of translation symmetry breaking order near 100 K, how it is related to the anomalous Hall effect, rotational symmetry breaking observed below about 100K. There have been proposals on a time-reversal breaking charge order, which can partially explain some experiments, such as the μ SR, Kerr rotation and (arguably) AHE.

The authors provide a different perspective based on their finding of the $d + id$ excitonic order due to twofold van Hove singularities. I think this is a well motivated toy model beyond the previously considered single band model.

Response: We thank Referee # 1 for their positive evaluation of the timeliness, novelty and motivation of our results.

Referee: *However, I have questions on how the new physics obtained here, i.e. $d+id$ excitonic order, provide new insights on understanding the puzzling experiments in AV_3Sb_5 compounds. My concerns are the following: First, there have been many experimental evidence for a translation symmetry breaking charge order near 100K, but it is not present in the phase diagram obtained by the authors, as the $d+id$ excitonic order carries zero momentum. Second, as far as I understand from the manuscript, the only new signature of the excitonic order that is not present with only charge order (time reversal even or odd) is the claim “but DFT calculations and magnetotransport indicate that the roughly hexagonal Fermi contours hosting the twofold van Hove singularities remain in the reconstructed band structure, which mainly undergoes changes near the K points [50]” (quoted from the IV Discussion section of the manuscript). However, from other studies, e.g. Ref. 1, the twofold van Hove band structure is reconstructed at M points. For this reason, to provide insights on the AV_3Sb_5 , I think the authors should make a more careful comparison with the charge ordering scenario, and explain to what extent, their findings can fit into the observation of translation symmetry breaking charge order. On the other hand, given the puzzles to reconcile various experiments in AV_3Sb_5 , I agree that different ideas are very valuable, and the excitonic order, if realized, would be very cool. One theoretical possibility that I can think of is to generalize the RG framework developed by the authors, and look for a possible fixed point with charge order as the leading instability, and the excitonic order as a subleading instability. To generate charge order, the authors may consider introducing some negative bare interactions, as suggested in Ref. 2.*

Response: Referee # 1 considers a more detailed discussion of the charge ordering in AV_3Sb_5 essential to accepting the manuscript. In response, we have both added extensive additional results analysing the interplay between excitonic order and charge order, and expanded the discussion of how our results related to experiment. In summary:

1. **We have taken bare interactions directly from first principles calculations for AV_3Sb_5 .** Additionally, as suggested by the referee, we additionally allow 3 of the 20 bare couplings to become zero or negative. The exact procedure and couplings are detailed in the Supplement (c.f. Tables S2 and S3).
2. **We derive the Landau-Ginzburg free energy for CDW and excitonic order.** We find the nontrivial result that these two phases coexist rather than compete.
3. **The resulting ground state features coexisting CDW and excitonic order.** Minimising the free energy allows us to show that the TvHS model predicts a scenario in which CDW appears as a high temperature ordered state in AV_3Sb_5 , which coexists with chiral excitons at lower temperatures.

Relating our scenario to experiment, we point out our phase diagram predicts that TRS breaks independently of translational symmetry, unlike in the complex CDW scenario. A smoking gun test to disentangle the two experimentally is to search for TRS breaking without translational symmetry breaking – for instance, by varying pressure strain or doping, one may suppress the CDW state while maintaining chiral excitons. Additionally, we anticipate the possibility of exciton signatures in tunnelling measurements, via Andreev tunneling. We we have added a more elaborate discussion of these points to the discussion section.

Also, we have removed the comment ‘the twofold van Hove singularities remain in the reconstructed band structure’, as it is misleading to the reader. We work at $T \lesssim 100$ K near the onset of CDW order, which implies a small CDW order parameter magnitude. In such a regime the role of CDW can be accurately captured perturbatively, i.e. via a Landau-Ginzburg free energy expansion. Note, on the other hand, for temperatures much less than 100 K the CDW magnitude becomes significant and it is then necessary to treat CDW nonperturbatively, i.e. to work with the effect of the CDW reconstructed bands. Hence, for comparison with AV_3Sb_5 , we truncate our calculations at $T = 50$ K; properties of the model at lower temperature may be explored in future work.

Referee: *Besides the major problem discussed above, below are some detailed questions and comments on the manuscript:*

1. In Eq. 5, the authors presented the pp and ph polarization bubble for intra and inter layer states with leading double logarithm. I have questions about the coefficients, are these for different channels really the same ($= \nu_0/4$)? At least, I think the Π_{pp}^{ii} and Π_{ph}^{ii} are different by a factor 1/2 (see Appendix B in Ref. 3). Naively, I would think d_1 and d_3 are different as well.

Response: We thank the referee for pointing out this typo – there should indeed be a relative factor of half, which can be deduced from a simple phase space argument. The revised Supplement treats the susceptibilities (and Landau-Ginzburg parameters) in some detail, plotting their evolution with temperature.

Referee: 2. *I think the manuscript should have more discussions on how the phase diagram depends on various parameters, such as the d_a coefficients introduced in Eq (5), the bare couplings. As far as I can tell, the phase diagram is not obtained from any fixed “point”, but from the running of coupling coefficients. I am worried that the phase diagram can be sensitive to bare parameters as it is not from RG fixed “point”. In particular, what is the true fixed point for the phase diagram shown in Fig. 2?*

Response: All three referees were interested in a more detailed analysis of the phase diagram. In our revised manuscript we have made the following changes:

1. **We provide a more extensive analysis of the possible fixed rays.** We present an exhaustive characterisation of the stable RG rays, as requested. We find chiral excitons appear naturally as a fixed ray for both kagome bilayers and honeycomb bilayers, with and without particle-hole symmetry. The fixed rays demonstrate excitonic order appears without fine-tuning, and are generic weak coupling instabilities.
2. **We provide estimates of the physically relevant bare parameters for AV_3Sb_5 .** Mapping the ab initio parameters of [Phys. Rev. Lett. **127**, 177001 (2021)] onto our interaction model, we establish an estimate for the 20 bare coupling values. By taking these bare values as input we compute the RG flow and resulting phase diagram. To demonstrate the proposed scenario for AV_3Sb_5 – our Figs. 4(a)ii [and (b)ii] – we have relaxed 2 [and 3] out of the 20 bare couplings.

Referee: 3. *The manuscript focus on log divergent van Hove singularities for both m and p bands. To apply to AV_3Sb_5 with greater accuracy, one of the band should be a quartic-saddle-point as the authors acknowledged. It would be very nice if there can be some discussions on how this could modify the phase diagram, even at a qualitative level.*

Response: Referee #2 was also interested in this question. We discuss in detail below. In summary, our argument is: **logarithmic RG is valid in the infrared, beneath a scale set by the quartic term in the dispersion.**

Expanding the electronic dispersion near a van Hove singularity, we may write $\varepsilon_{\delta k} = a_2(\delta k)^2 + a_4(\delta k)^4$, where δk is the momentum deviation from the saddle point. A pure higher-order vHS corresponds to $a_2 = 0, a_4 \neq 0$. When considering AV_3Sb_5 , e.g. [Nat. Phys. **18**, 301–308 (2022)], the presence of $a_4 \neq 0$ has been referred to as a ‘higher-order vHS’. However, in that work (and supported by ab initio calculations [Phys. Rev. Lett. **127**, 177001 (2021)]), there still exists $a_2 \neq 0$.

From the perspective of RG, the presence of $a_2 \neq 0$ implies that in the infrared the susceptibilities ultimately exhibit a logarithmic divergence. For ‘pure’ higher-order vHS ($a_2 = 0, a_4 \neq 0$), the flow becomes power-law divergent, and requires a modification of our analysis. With the assumption of a nonzero a_2 , the quartic corrections to the dispersion become subdominant to the non zero quadratic term beneath an energy scale $\varepsilon_{HO} \simeq 2a_2^2/a_4$.

To support this assumption, we refer to the Supplementary Material of Nat. Phys. **18**, 301–308 (2022) in which a fit to ARPES data shows $a_2 = 1.2 \text{ eV}\text{\AA}^2$, $a_4 = 12 \text{ eV}\text{\AA}^4$, and see the energy scale $\varepsilon_{HO} = 0.24 \text{ eV}$, below which the quartic term becomes irrelevant. Importantly, our theory has an ultraviolet cut-off at $\Lambda_{UV} \sim \varepsilon_{HO} = 0.5 \text{ eV}$, and subsequently flows to an infrared scale set by the temperature $\Lambda_{IR} = T$. We have $\Lambda_{IR} \ll \varepsilon_{HO}$, and therefore a majority of the RG flow is within the regime of logarithmic scaling; specifically, for the results of Fig. 4, $T \lesssim 100 \text{ K}$. With these remarks in mind, we are confident that quartic corrections to the dispersion should not significantly modify our results.

Point-by-point response to the comments made by Referee # 2:

Referee: *The manuscript presents a model study of a system of interaction fermions on a hexagonal lattice geometry featuring two active electronic bands near van-Hove filling where the two bands exhibit opposite concavity – a configuration which has been coined “two-fold van-Hove singularity”. This specific configuration facilitates the formation of a d-wave excitonic condensate, which the authors argue to be a topologically non-trivial state of matter, but also unconventional superconductivity. The model construction is inspired by recent experiments on kagome metals, which found evidence for unconventional superconductivity and time-reversal symmetry breaking.*

To investigate the correlated states of their model and their competition, the authors employ a standard parquet RG and they also carry out Landau-Ginzburg analysis for the interesting excitonic condensate state, which is argued to feature a quantum anomalous Hall conductance. They further discuss the response of the system to variations of the filling and effective interaction parameters.

The manuscript is generally well written, and well-structured and it is overall a pleasant read. The considered problem is well introduced and properly motivated. The work is technically involved, mainly because of the large number of couplings in the RG approach, but at the same time it is methodically quite straightforward employing the well-established parquet RG. The concomitant technicalities of the method are introduced in a self-contained way with ample details postponed to the appendices. References and the concluding discussion are appropriate. Also, I find the essential results of the manuscript and the general scenario very interesting and I believe that the work may provide significant impact.

Response: We thank the referee for their assessment that our work is well-motivated, carefully analysed, readable and potentially significantly impactful.

Referee: *I have, however, a major reservation when it comes to recommending the work for publication in Nature Communications. In my opinion, the presentation of the work is rather technical, i.e. I would consider the general style to be exceedingly appropriate for example for the APS Physical Review journals. For Nature Communications, on the other hand, I would expect the essential physical results to be presented in a manner that is more accessible to its broad readership. In particular, I think that the discussion of the interesting topological d-wave excitonic condensate state and its underlying physical mechanism deserves to be more clearly exposed in the manuscript, which I believe is possible without resorting to an overly technical discussion. So, my central point of criticism is that the manuscript, if considered for Nature Communications, requires a major revision concerning the presentation of the results according to the principle indicated above.*

Response: We agree with Referee #2 that the manuscript would be more suitable for Nature Communications if the presentation were substantially revised. We have therefore made several revisions to the section on the RG approach and general style.

The manuscript has undergone major revisions, with a focus on emphasising key qualitative features and elucidating the physical mechanisms underpinning chiral excitons and CDW in a more accessible format. We have kept a high-level discussion of the methods/techniques in the main text, and relegated explicit expressions for the interactions and gap equations to the Supplement, with some key details in a Methods section. The main text now predominantly focuses on the key concepts: that the opposite concavity vHS result in new excitonic phases, which we analyse in an accessible way using Landau-Ginzburg theory.

We believe the level of technical details is now commensurate with other theory papers published in Nature Communications.

Referee: *I append two additional comments/questions that should be addressed in a revised version of the manuscript, independent of the publication venue: (1) One important shortcoming of the presented approach is that it does not allow for a real material-specific prediction about the leading Fermi-surface instability, because it only resolves the very low-energy excitations around the saddle-points of the energy dispersion. Therefore, no realistic Coulomb interactions can be taken into account. Instead, the authors have to choose more or less arbitrary initial values for their couplings g , h , j , l . Nevertheless, it is known that the strength and range of the Coulomb interactions can have a profound impact on the ground state. The parquet RG reflects this fact partly by featuring several RG fixed trajectories which are realized for different initial combinations of the couplings g , h , j , l . The authors have only considered a very small subspace of the corresponding parameter space, potentially disregarding a very large amount of interesting correlated states. I think that it would be interesting to provide a more exhaustive account for the possible RG fixed trajectories and their concomitant correlated states.*

Response: We agree with Referee #2 that the treatment of the phase diagram in our original submission was not sufficiently exhaustive – we have discussed this issue in our responses to Referee #1.

To respond to Referee #2 a bit more directly, we have indeed provided an exhaustive account of the possible RG fixed rays and the resulting ordered phases. This analysis is carried out for three variants of our TVHS model. Crucially, this RG fixed ray analysis demonstrates that chiral excitons are a universal feature of the TvHS model, as fixed rays with chiral excitons appear in all three variants. Additionally, we have made stronger contact with material-specific physics by investigating the RG flow resulting from initial couplings motivated by first principles calculations. This analysis is non-universal, but is very important to demonstrate our proposed scenario for the AV_3Sb_5 – that CDW and chiral excitons coexist, and that may appear as relatively higher temperature phases $T \sim 100$ K. Coexistence is a robust property of the model, arising from the Landau-Ginzburg free energy, whereas the critical temperature depends on chosen input parameters.

Referee: (2) *The authors mention that the ARPES data on the vanadium metals seem to feature a higher-order van-Hove singularity (due to the quartic saddle point) at least in one of the bands. This is a very interesting remark, but at the same time it calls the applicability of the presented results on the specific material into questions. As has been studied elsewhere, the presence of higher-order van-Hove singularities substantially changes the hierarchy of leading contributions to the RG evolution equations and therefore also alters the leading Fermi-surface instabilities. I think that the authors need to provide an argument on why this fact does not fundamentally change their results or they need to expose more clearly that they are not obviously applicable to the vanadium metals.*

Response: Referee #1 was similarly interested in how a higher-order vHS might alter our results. As stated in our responses to Referee #1, the argument for why the higher vHS does not change our results is essentially that the energies scales we are interested in for AV_3Sb_5 exceed the energy scale at which ARPES data suggest the RG equations would be modified by the higher order vHS.

Despite there being a non-negligible *quartic* correction (associated with a higher-order vHS), as indicated by experiment [Nat. Phys. **18**, 301–308 (2022)], the same experiment as well as ab initio calculations [Phys. Rev. Lett. **127**, 177001 (2021)] show a non-zero *quadratic* term near the M -point. Ultimately, below an energy scale ε_{HO} this quadratic term dominates. We estimate this scale to be $\varepsilon_{HO} \simeq 0.25$ eV. We conclude that the infrared physics of interest in our work, i.e. below $T_c \simeq 100$ K, is safely within the *quadratic* regime. Technically, this means that the logarithmic RG procedure employed is well justified.

Point-by-point response to the comments made by Referee # 3:

Referee: *The manuscript “Chiral excitonic order, quantum anomalous Hall effect, and superconductivity from twofold van Hove singularities in kagome metals” is a numerical study of the leading electronic instabilities in a model inspired by the recently discovered class of kagome metals. The authors use perturbative renormalization group analysis and focus on the effect of the multiple van Hove singularities present in the multi-orbital kagome low-energy model, put forward previously for these compounds. The topic is timely and the results reported are novel. The finding of an excitonic d-wave order parameter and the connection to quantum anomalous Hall response have the potential to become relevant in the fast growing field of kagome metals.*

Response: We thank the referee for their assessment that our work is timely, novel, and relevant.

Referee: *However, before supporting publication on Nature Communications, I would like the following points to be more elaborated on by the authors: 1) Choice of interaction: even if the model on which the authors focus is highly idealized, the importance of this study strongly relies on its connection to the realistic compounds. For this reason, the choice of the multi-flavor interaction in Eq. (3) must – at least qualitatively – be motivated by a first-principle calculation. Recently the first cRPA calculation have started to appear in the literature [see e.g. <https://doi.org/10.48550/arXiv.2203.05038>] so the authors can connect to the values reported there. In this respect, I suggest adding a discussion on the choice of the coupling strength in the flavor basis (not in the band basis) as well as a check of the robustness of the main outcome discussed, against changes of these coupling strengths in the ballpark of the cRPA values.*

Response: We would also like to thank Referee #3 for the suggestion to employ ab initio cRPA results. And agree this would allow us to make stronger connection to experiment. To this end, we have taken the model parameters established in [Phys. Rev. Lett. **127**, 177001 (2021)], which are based on cRPA. These parameters are used directly to estimate the bare coupling constants in our effective model, as shown by Table I in the revised manuscript. We allow some relaxation of these explicit parameters, as discussed in the main text and summarised in Tables S2 and S3 of the Supplement. The resulting phase diagrams are presented in Fig. 4(a)ii and (b)ii.

Referee: 2) *The authors stress the opposite concavities of the van Hove singularities but do not consider that the “pure” and “mixed” ones differ also for their spatial support. As discussed in Ref. 9, 8 and 11, and shown in Ref. 53 (see Fig. 3e-g of Ref. 53) the decisive difference between them is their complementary sublattice localization (from which their names). The Bloch wave function of the pure van Hove localizes only on one sublattice (more precisely, the charge profile at each of the three M points has a maximum on one of the three sublattice of the kagome unit cell) while the Bloch wave function of the mixed van Hove localizes on the remaining two sublattices, respectively. This important distinction, first discussed in Refs 9, 8, and 11 and recently illustrated and supported by measurements in Ref. 53 has to be explained in the present manuscript, otherwise the only distinction seems to be the hole- or electron-like character of these bands. An extremely interesting thing would be to explicitly analyze the effect of this spatial support on the different types of instabilities discussed here, given the fact that pure and mixed van Hoves are believed to react different to local and non-local Coulomb terms. One further relevant point is to analyze the relative importance of the concavity compared to the spatial support of these bands for the d-wave excitonic order. The manuscript would profit strongly if a discussion on these points, based on the numerical analysis of the authors, could be added. Moreover, at the end of section II B, the above sublattice support should be introduced as important property of the kagome van Hove bands, in addition to their quadratic/quartic momentum dispersion.*

Response: We agree with Referee #3 that this is a very important point; in fact several experimentalists we have spoken to asked about it. We have made this issue a central feature of the new results added to the manuscript.

In order to incorporate the differences between p -type and m -type vHS, it becomes necessary to accommodate different intra-flavour couplings g_{ic} and g_{id} , extending the previous 16 coupling model to a 20 coupling model. We discuss the results for both models – the 16 coupling model is primarily relevant to honeycomb systems, where there is no sublattice interference effect, and the 20 coupling model is relevant to kagome system where the m -type and p -type distinction becomes important. As the referee predicted, the different spatial support of the band wavefunctions causes some couplings to vanish; we find that this has important consequences for the RG flow. When estimating the bare values of the couplings from [Phys. Rev. Lett. **127**, 177001 (2021)], we project the interaction parameters onto the m -type and p -type band wavefunctions, which explicitly demonstrates this effect.

Fig. 2 provides a universal set of weak-coupling leading phases, as obtained from the RG fixed rays, for both the 16 and 20 coupling models, whereas we exclusively use the 20 coupling model when proposing the phase diagram for AV_3Sb_5 , i.e. Fig. 4.

Referee #3 also suggests “Moreover, at the end of section II B, the above sublattice support should be introduced as important property of the kagome van Hove bands, in addition to their quadratic/quartic momentum dispersion.” We have followed their recommendation, and added additional discussion of this point at the end of Section II B.

Referee: *Minor points: -in Fig. 1b it could be more explicitly indicated that the upper band comes from the kagome because, given the restricted y-range, one does not see the flat band. Of course, looking at the sign of the parameter γ_0 one can get this information but it is not immediately direct for a reader.*

Response: We agree that this benefits from clarification. The plot shows bandstructure for a two-flavour kagome model in the resubmission, but we have now colour-coded the plot to indicate the orbital content of the bands.

Referee: *The phase diagrams in Fig. 2 seem really sketched by hand rather than been drawn on the basis of some results. The authors write ”we arrive the the phase diagram Figure 2.” (BTW, watch out the typo) but I guess some numbers at least could be added to give these plots a bit of quantitative look*

Response: We thank the referee for calling our attention to the typo, and have replaced this phase diagram with extensive new results: the universal fixed ray analysis [Fig. 2], and the quantitative phase diagrams guided by ab initio calculations [Fig. 4].

REVIEWER COMMENTS

Reviewer #2 (Remarks to the Author):

In the second version of their manuscript Scammell et al. have include major revisions as compared to their first version. These revisions clearly reflect the comments by the referees, and they appear to be appropriate. Also, the style of presentation has improved considerably, which I criticized in my first report as referee 2 and it is now much better suited for Nature Communications.

Moreover, the authors have also addressed my physics related questions on their manuscript. They have added a discussion on the choice of the Coulomb interactions, which define a starting point of their renormalization group analysis attempting to connect more directly to ab initio data and experiments. They then use these initial interactions to calculate the newly included phase diagram in Fig. 4.

I have doubts that the presented strategy to choose initial conditions for the renormalization group flow is appropriate. The reason is the following: In my understanding the initial interactions from the cRPA method are a reasonable estimate at the scale of the extended Hubbard model for the investigated materials, i.e., the full band structure from the tight-binding part plus the short-ranged Coulomb interactions. The initial cut-off scale chosen by the authors is much smaller than the bandwidth, because it must accommodate the saddle-point behavior of the dispersion near the van-Hove points. It can be expected that fluctuations between the scale of the bandwidth and the cut-off scale chosen by the authors severely modify the profile of the interactions, i.e., the ratios of U , V , J' , etc. Hence a reasonable starting point for the initial interactions would include such fluctuations before calculating the RG trajectories leading to the phase diagram in Fig. 4. Therefore, I believe that it is questionable whether these phase diagrams can tell a lot about the material, and I think that the exhaustive analysis of the universal RG behavior is the best one can get from this approach without further input.

For the above reason, I cannot support publication of the manuscript in the present form, however, I am willing to reconsider, if the authors can clarify this point and revise their work accordingly.

Reviewer #3 (Remarks to the Author):

The authors have addressed all my points and in my opinion the new version is ready to be published in Nature Commun.

Resubmission of NCOMMS-22-03563-A: “Chiral excitonic order from twofold van Hove singularities in kagome metals”

Referee 1: It has been acknowledged that the requirements for publication, set by Referee 1, have been met.

Referee 3: *The authors have addressed all my points and in my opinion the new version is ready to be published in Nature Commun.*

Point-by-point response to referee #2:

Referee: *In the second version of their manuscript Scammel et al. have included major revisions as compared to their first version. These revisions clearly reflect the comments by the referees, and they appear to be appropriate. Also, the style of presentation has improved considerably, which I criticized in my first report as referee 2 and it is now much better suited for Nature Communications. Moreover, the authors have also addressed my physics related questions on their manuscript.*

Response: We thank the referee for their positive assessment.

Referee: *They have added a discussion on the choice of the Coulomb interactions, which define a starting point of their renormalization group analysis attempting to connect more directly to *ab initio* data and experiments. They then use these initial interactions to calculate the newly included phase diagram in Fig. 4. I have doubts that the presented strategy to choose initial conditions for the renormalization group flow is appropriate. The reason is the following: In my understanding the initial interactions from the cRPA method are a reasonable estimate at the scale of the extended Hubbard model for the investigated materials, i.e., the full band structure from the tight-binding part plus the short-ranged Coulomb interactions. The initial cut-off scale chosen by the authors is much smaller than the bandwidth, because it must accommodate the saddle-point behavior of the dispersion near the van-Hove points. It can be expected that fluctuations between the scale of the bandwidth and the cut-off scale chosen by the authors severely modify the profile of the interactions, i.e., the ratios of U , V , J' , etc. Hence a reasonable starting point for the initial interactions would include such fluctuations before calculating the RG trajectories leading to the phase diagram in Fig. 4. Therefore, I believe that it is questionable whether these phase diagrams can tell a lot about the material, and I think that the exhaustive analysis of the universal RG behavior is the best one can get from this approach without further input. For the above reason, I cannot support publication of the manuscript in the present form, however, I am willing to reconsider, if the authors can clarify this point and revise their work accordingly.*

Response:

We firstly clarify that the phase diagrams of Fig. 4(a)i and 4(b)i are not reliant on *ab initio* calculations. In these phase diagrams, we treated the interaction eigenvalues (which are linear combinations of the coupling constants) as undetermined free parameters, with magnitudes spanning a range that includes those of Table I. The purpose of these phase diagrams is to illustrate the universal feature of the Landau-Ginzburg free energy – that there is a broad range of couplings within which CDW and excitons coexist.

Regarding Figs 4(a)ii and 4(b)ii, the referee raises a good point; the input parameters are taken at the lattice scale, whereas our patch RG treatment begins at larger length scales. We agree that there is some running of the couplings between these two scales, and that this is not accounted in our computation of Figs 4(a)ii and 4(b)ii. We stress that this assumption is only used for the results of Figs 4(a)ii and 4(b)ii.

We believe that although Figs 4(a)ii and 4(b)ii should not be understood as quantitative, these two sub-figures still illustrate a scientifically valuable point – demonstrating that there exist physically reasonable bare values which produce the desired phenomenology. Taking values comparable to those found in *ab initio* calculations allows us to illustrate a physically reasonable avenue to a phase diagram with CDW at high temperatures and coexisting excitons at lower temperature. Indeed the fact that there are two phase di-

agrams here illustrating both parameters with rCDW and iCDW highlights how these plots are attempting to illustrate a scenario, rather than accurately microscopically model AV_3Sb_5 . We recognise that a more precise estimation of these initial values goes beyond our analysis and certainly warrants further investigation – for instance, a future study using fRG could compliment our analysis.

In the revised manuscript, we make explicit the shortcoming in our approximation – first under Table I and then again at the end of the Results section. In particular, we stress in the revised manuscript that these two sub-figures should not be taken as quantitative predictions for AV_3Sb_5 , but as physically motivated illustrations of the CDW/chiral exciton scenario. We believe that this adequately informs the reader about the quantitative accuracy of the initial conditions.

Yours sincerely,

Harley Scammell and Julian Ingham

REVIEWERS' COMMENTS

Reviewer #2 (Remarks to the Author):

The authors have appropriately addressed the criticism that I expressed in my last report and I can now support publication in Nature Commun.